# Structural absorption by barbule microstructures of super black bird of paradise feathers

Dakota E. McCoy [1], Teresa Feo[2], Todd Alan Harvey[3] & Richard O. Prum[3]

Many studies have shown how pigments and internal nanostructures generate color in nature. External surface structures can also influence appearance, such as by causing multiple scattering of light (structural absorption) to produce a velvety, super black appearance. Here we show that feathers from five species of birds of paradise (Aves: Paradisaeidae) structurally absorb incident light to produce extremely low-reflectance, super black plumages. Directional reflectance of these feathers (0.05–0.31%) approaches that of man-made ultra-absorbent materials. SEM, nano-CT, and ray-tracing simulations show that super black feathers have tilted arrays of highly modified barbules, which cause more multiple scattering, resulting in more structural absorption, than normal black feathers. Super black feathers have an extreme directional reflectance bias and appear darkest when viewed from the distal direction. We hypothesize that structurally absorbing, super black plumage evolved through sensory bias to enhance the perceived brilliance of adjacent color patches during courtship display.

[1] Department of Organismic and Evolutionary Biology, Harvard University, Cambridge, MA 02138, USA. [2] Department of Vertebrate Zoology, MRC-116, National Museum of Natural History, Smithsonian Institution, Washington, DC 20013, USA. [3] Department of Ecology and Evolutionary Biology, and Peabody Museum of Natural History, Yale University, New Haven, CT 06520, USA. Dakota E. McCoy and Teresa Feo contributed equally to this work. Correspondence and requests for materials should be addressed to D.E.M. (email: dakotamccoy@g.harvard.edu)

**B**ird coloration is a model system for understanding evolution, speciation, and sexual selection[1]. Color-producing mechanisms are generally assigned to two categories[1]: (i) pigmentary colors produced by molecules and (ii) structural colors produced by light scattering from nanoscale variation in refractive index (e.g., channels of air within a keratin matrix). In addition to color, the directional distribution of scattered light can also affect plumage appearance. The shape, orientation, and smoothness of the feather barbs and barbules create directionally dependent appearance, such as with glossy or iridescent plumage[2, 3].

However, the mechanism of "structural absorption[4–8]," which occurs when superficial features cause multiple scattering of light[5, 9], can also influence visual appearance. Each time light scatters at a surface interface, a proportion of that light is transmitted into the material, where it can be absorbed[9]. By increasing the number of times light scatters, structurally absorbing materials can increase total light absorption to produce a profoundly black appearance. For example, a shiny metal with a smooth surface that reflects 30–70% of visible light can be converted to a matte black material that reflects less than 5% of light by adding microstructural surface complexity that increases structural absorption[5]. Natural examples of structural absorption have been described in the wing scales of butterflies[10–12] and the body scales of a snake[13]. Structurally absorbing, "super black"[14, 15] materials (which have extremely low, broadband reflectance) have important applications for a wide range of optical, thermal, mechanical, and solar technologies, including thin solar cells[4] and the lining of space telescopes[8].

Decades of previous research have focused on the physics, chemistry, social function, and evolutionary history of bird plumage coloration[1, 16]. The polygynous birds of paradise (Aves: Paradisaeidae) have evolved some of the most elaborate mating displays and plumage ornaments in all animals[17] (Fig. 1). In multiple species from multiple genera in the family, males have deep, black, and velvety plumage patches immediately adjacent to brightly colored, highly saturated, and structurally colored plumage patches (Fig. 1c–g). These black plumage patches have a strikingly matte appearance (i.e., lacking specular highlights) and appear profoundly darker than normal black plumage of closely related species[18] (Fig. 1a, b).

Here we use spectrophotometry, scanning electron microscopy (SEM), high-resolution synchrotron tomography (nano-CT), and optical ray-tracing simulations to investigate the role of structural absorption in black feathers from seven species of birds of paradise. Unlike normal black feathers with typical barbules, we find that super black feathers have highly modified barbules arranged in vertically tilted arrays, which increase multiple scattering of light and thus structural absorption. Super black feathers reduce specular reflection by one to two orders of magnitude compared to that of normal black feathers and have extreme directional bias corresponding to the viewing direction of a female observing a displaying male. Therefore, we hypothesize that these feathers evolved to enhance the perceived brilliance of adjacent color patches by generating an optical/sensory illusion during mating displays.

## Results

**Reflectance spectra.** We visually selected five species of polygynous birds of paradise with profoundly black plumage from five different genera—*Ptiloris paradiseus, Seleucidis melanoleucus, Astrapia stephaniae, Lophorina superba*, and *Parotia wahnesi*—and two species with normal black plumage—*Lycocorax pyrrhopterus* and *Melampitta lugubris* (a Papuan corvoid related to birds of paradise)—to serve as comparative controls

(Supplementary Table 1). For *Lophorina*, we examined both the profoundly black plumage of the display cape and the normal black plumage of the back, which is not used in display.

We measured the spectral reflectance of each plumage patch using two methods: (1) total integrated (specular and diffuse) reflectance was measured using an integrating sphere with a diffuse light source, and (2) normal directional reflectance was measured with a directional light source and a detector oriented normal to the feather vane (see "Methods" for details). Both the total integrated and normal directional reflectance measurements confirmed that the profoundly black plumage patches were darker than normal black plumage (Fig. 2, Supplementary Figs. 1 and 2, and Supplementary Table 1). The directional reflectance of the profoundly black plumage patches was extremely low (0.05–0.31%), and was one to two orders of magnitude less than the normal black plumages (3.2–4.7%) (Fig. 2b, Supplementary Fig. 2, and Supplementary Table 1). The extremely low directional reflectance of these five plumages is comparable to that of other natural and man-made super black materials[4–8, 10–15].

Reflectance spectra differed between normal and super black plumage: spectra of normal black plumages sloped upward above ~600 nm (Fig. 2 and Supplementary Figs. 1a–c and 2a–c), which is typical of melanin pigments[19]. In contrast, reflectance spectra of all five super black plumages were nearly flat (Fig. 2 and Supplementary Figs. 1d–h and 2d–h), which is reminiscent of super black carbon nanotube materials with exceptionally low reflectance over the entire visible range[14]. Super black plumage reflectance curves were also flatter than many man-made velvet fabrics (Supplementary Fig. 3), profoundly black snake scales[13], and butterfly scales[12]. Super black plumages of birds of paradise appear to have a more efficient, broadband absorption than other biological examples of structural absorption.

**Feather microstructure.** SEM and nano-CT revealed striking differences in microscopic barbule morphology between normal black and super black feathers (Fig. 3a, b and Supplementary Figs. 4–6). Barbules of normal black feathers had a typical, open pennaceous morphology with smooth margins and a horizontal orientation restricted to the plane of the barb rami (Fig. 3a and Supplementary Figs. 4a, b, 5a, b, and 6a, b). In contrast, barbules of all super black feathers had a highly modified morphology, with microscale spikes along the margins, reminiscent of dried oak leaves. Super black barbules curved up from the plane of the barb rami to form a densely packed array tilted ~30° toward the distal tip of the feather (Fig. 3b and Supplementary Figs. 4c–g, 5c–f, and 6c–f). The resulting morphology—an array of deep, curved cavities between the smallest branches of the feather vane—is distinct from the microstructure of super black snake and butterfly scales[13] and from man-made super black materials[4–8].

Often, color-producing feather pigments or nanostructures are restricted to the exposed tips of overlapping feathers in the plumage[1]. We found a similar pattern with super black barbule modifications. Barbules toward the tip of super black feathers were highly modified, whereas barbules toward the base of the same feathers had a typical normal morphology (Supplementary Fig. 7). Also, black feathers from the back of *Lophorina superba*, which are not used during display, had a typical normal morphology and were more reflective than super black feathers from the display cape with modified barbules (Supplementary Figs. 1c, h and 2c, h and Supplementary Table 1). These observations support the conclusion that the modified barbule morphology of super black feathers serves an optical, signaling function.

Structural absorption can occur when superficial cavities that are much greater in width than the wavelength of visible light

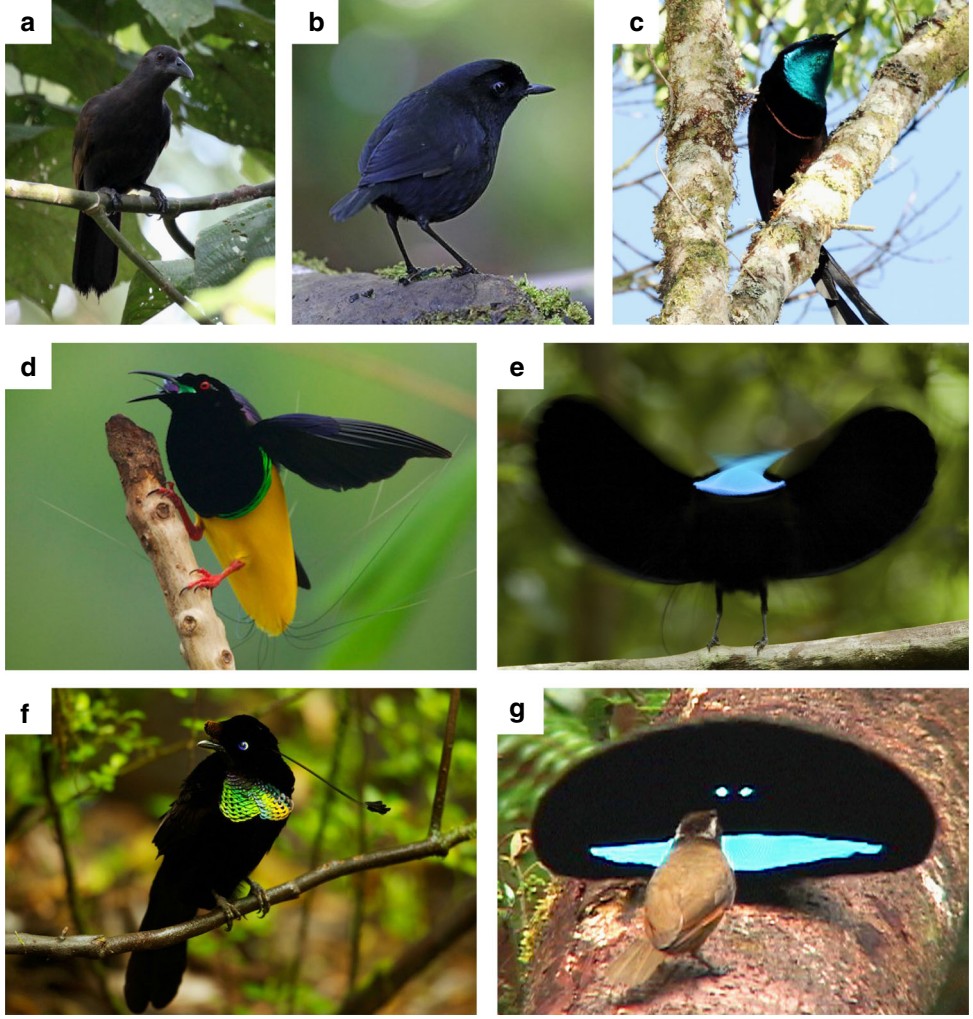

**Fig. 1** Six species of birds of paradise and one close relative. **a**, **b** Species with normal black plumage patches. **c**–**g** Species with super black plumage patches. **a** Paradise-crow *Lycocorax pyrrhopterus*. **b** Lesser Melampitta *Melampitta lugubris*, a Papuan corvoid closely related to birds of paradise. **c** Princess Stephanie's Astrapia *Astrapia stephaniae*. **d** Twelve-wired Birds-of-Paradise *Seleucidis melanoleucus*. **e** Paradise Riflebird *Ptiloris paradiseus* during courtship display. **f** Wahnes' Parotia *Parotia wahnesi*. **g** Superb Bird-of-Paradise *Lophorina superba* during courtship display with female (brown plumage). Photo credits: **a** @Hanom Bashari/Burung Indonesia; **b** Daniel López-Velasco; **c** Trans Niugini Tours; **d**–**f** Tim Laman; **g** Ed Scholes

cause multiple scattering of light[5]. Even shiny metal surfaces can appear black if they have the appropriate surface microstructure[11–13]. The tilted barbule arrays in super black bird of paradise feathers had intra-barbule cavities that were ~200–400-μm deep and ~5–30-μm wide, with smaller cavities along the barbule margins at a < 5-μm scale (Fig. 3b and Supplementary Figs. 4–6). Remarkably, the super black feathers retained their velvety black appearance even after sputter coating with gold for SEM, whereas the normal black feathers appeared gold (Fig. 3c, d). This direct, experimental evidence shows that super black feathers structurally absorb light to create their profoundly dark appearance.

**Light-scattering simulations**. To directly quantify the effects of barbule surface microstructure on light absorption in feathers, we used virtual ray-tracing simulations to model the interaction of light with 3D nanoscale tomographic models of normal black and super black feathers (Supplementary Fig. 8). Ray-tracing simulations calculate the path and radiant power of light rays as they interact with a 3D model. Each time a simulated light ray intersects the feather surface (a scattering event), a portion of its radiant power is reflected from the surface, and the remaining portion is transmitted into the material where it can be absorbed. Our simulations assumed no surface roughness and 100% absorption of transmitted light. These assumptions restricted light scattering to the specular direction and allowed us to control for any variation in pigmentation, internal structure, or surface roughness that might be present in the real feathers. Thus, the ray-tracing experiments isolated the effects of external feather microstructure on light scattering to characterize structural absorption among feathers with different barbule morphologies.

First, we conducted ray-tracing simulations that modeled the normal directional reflectance spectrophotometry measurements. Our simulations confirmed that feather barbule microstructure causes multiple scattering of light (Fig. 4a and Supplementary Table 1). The percentage of light rays that scattered at least twice varied among feathers from 33 to 95%, documenting the contribution of barbule morphology to light-scattering behavior (Fig. 4b and Supplementary Table 1). Super black feathers with modified barbule arrays caused more multiple scattering, and had greater simulated structural absorption, than normal black feathers (Fig. 4b and Supplementary Table 1). Furthermore, we found a significant negative relationship between measured normal, directional reflectance and the percentage of simulated

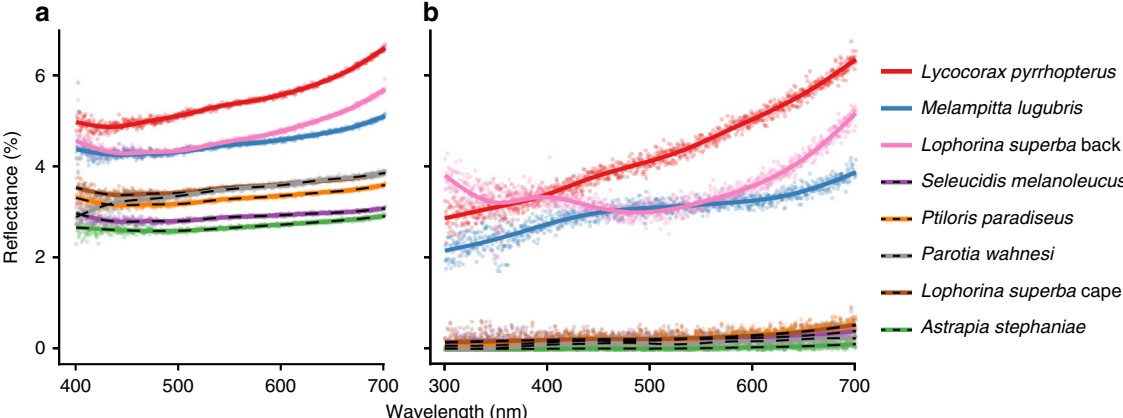

**Fig. 2** Reflectance spectra of black and super black plumages. **a** Total integrated (diffuse and specular) reflectance. **b** Normal, directional reflectance. Dotted lines are super black plumages. See Supplementary Figs. 1 and 2 for detailed spectra for each species

light rays that scattered at least twice (Fig. 4c; linear regression: $R^2 = 0.68$, slope = −0.063, SE = 0.021, and $P < 0.05$). These results demonstrate that the modified barbule arrays of super black feathers increase multiple scattering of light, and contribute to a darker appearance (i.e., lower reflectance) through increased structural absorption, relative to the typical barbule morphology of normal black feathers.

Next, we configured ray-tracing simulations with several different idealized lighting conditions to investigate how appearance varies with viewing direction and illumination. Total integrated reflectance measurements of super black feathers were only 50% lower than for normal black feathers (Fig. 2), indicating specular reflectance from other angles. Furthermore, the curved, laminar barbules of the super black feathers angle toward the distal tip of the feather, rather than projecting up perpendicular to the plane of the feather vane. This titled barbule orientation could produce directional variation in structural absorption, and thus reflectance[3]. To investigate, we calculated the simulated directional reflectance for four different lighting setups: (i) omnidirectional light, (ii) directional light tilted + 45° toward the proximal end of the feather, (iii) directional light at 0° normal to the feather (as above), and (iv) directional light tilted at −45° toward the distal end of the feather (see Methods for details). The omni-directional illumination (setup i) is comparable to light in an open environment on a cloudy day, whereas the directional illuminations (setups ii–iv) are comparable to light in a closed environment, such as the forest floor, with breaks in the canopy that constrain incident light to a narrow angular range[20].

Normal black and super black feathers differed markedly in their directional reflectance. Normal black feathers reflected light in a manner consistent with classical glossy surface reflection theory[21]: the majority of energy was reflected in directions roughly equal and opposite to that of the directional incident light (Fig. 5a and Supplementary Fig. 9a, b). Thus, the darkest viewing quadrant varied with the angle of illumination for normal black feathers (Supplementary Table 2). In contrast, super black feathers always reflected the majority of energy toward the proximal viewing quadrant, regardless of the angle of illumination (Fig. 5b and Supplementary Fig. 9c–f). Super black feathers were the darkest when viewed from the distal viewing angle (Supplementary Table 2), which corresponds to looking into the openings of the deep cavities between barbule tips (Fig. 5b and Supplementary Fig. 9).

## Discussion

Our findings demonstrate that super black bird of paradise feathers structurally absorbs up to 99.95% of directly incident light, and that variation in external surface microstructure can contribute to observed differences in visual appearance of bird plumage. The vertically tilted barbule arrays of super black bird of paradise feathers create deep, curved cavities. This morphology is distinct from the longitudinal ridges of butterfly scales[11] and the vertical cones of snake scales[13], substantially expanding the diversity of structurally absorbing biological materials in nature.

The extreme directional reflectance bias in super black feathers is congruent with field observations of bird of paradise courtship behavior[22]. Males of many species perform displays that maintain a specific directional orientation between their ornaments and the viewing females[17] (Fig. 1g). We hypothesize that the tilted barbule arrays function in coordination with the behavioral repertoire to ensure that females view super black plumage patches at their darkest orientation.

Interestingly, in both butterflies and birds of paradise, super black patches are always adjacent to bright, highly saturated, and structural colors. For example, Lophorina has a super black plumage display cape surrounding its intensely brilliant blue patches, but normal black plumage on the back that is not featured during display (Fig. 1g and Supplementary Figs. 1c, h and 2c, h). We hypothesize that structurally absorbing super black patches evolve because they exaggerate the perceived brilliance of adjacent color patches through a sensory/cognitive bias inherent in the vertebrate mechanism of color correction. Vertebrates use specular highlights, or white reflectance from object surfaces, within the visual field to correct for the spectrum and quantity of ambient light[23]. We propose that structurally absorbing super black patches (i) eliminate specular reflectances around the brilliant color patch, (ii) lower the observers perceived estimate of the quantity of ambient light upon that portion of the visual scene, and thus (iii) disrupt the perceiver's capacity to estimate the brilliance of the color patch. If the brain perceives that more light is coming from a patch than it estimates is ambient upon it, the patch will appear to be self-luminous or to float in space[24–27]. Perceptual experiments demonstrate this bias in the color correction mechanisms of goldfish (which are tetrachromats like birds) and humans[28].

Displays of some bird of paradise, like male Lophorina, produce exactly this self-luminous effect on human observers and in videos and photographs (Fig. 1g), and we predict that these plumages produce similar perceptual effects on avian observers. Theoretically, white balancing (i.e., von Kries correction) involves dividing the signal stimulus from each cone type by that cone

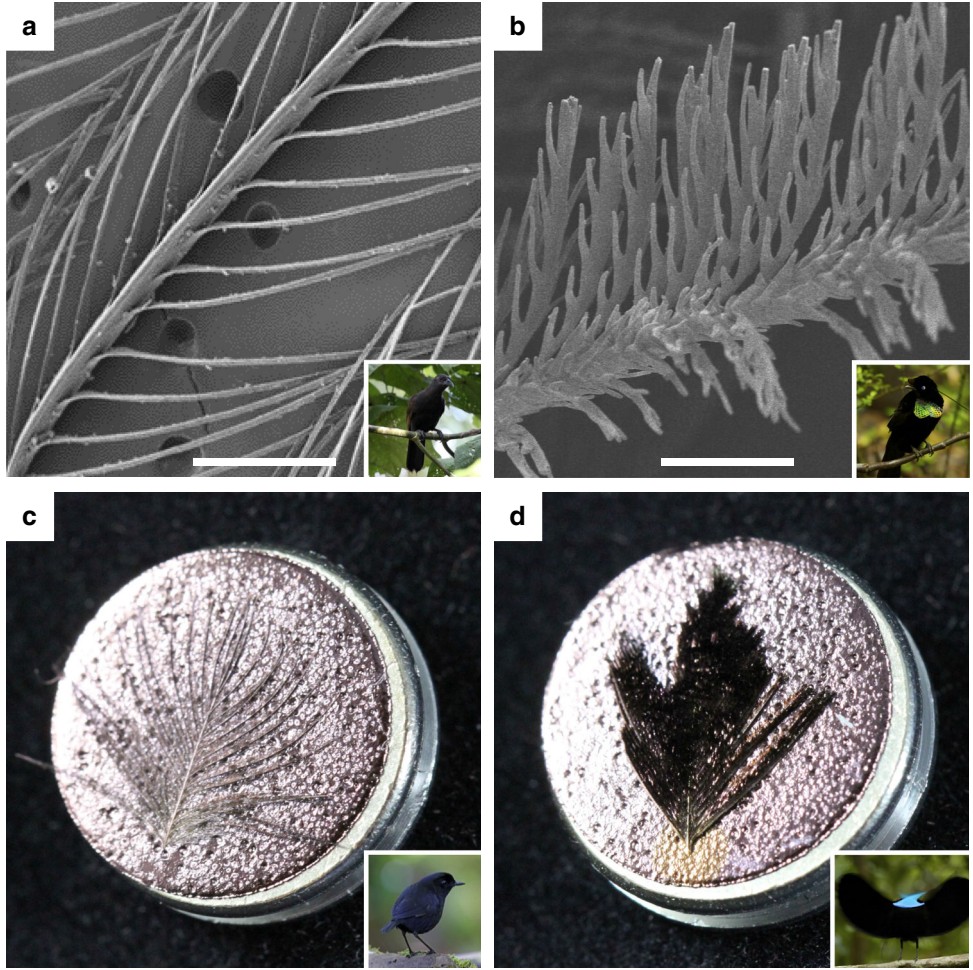

**Fig. 3** Examples of normal and super black feather microstructure. **a** SEM micrograph of *Lycocorax pyrrhopterus* normal black feather with typical barbule morphology; scale bar, 200 μm. **b** SEM micrograph of *Parotia wahnesi* super black feather with modified barbule arrays; scale bar, 50 μm. **c** Gold sputter-coated normal black breast feather of *Melampitta lugubris* appears gold. **d** Gold sputter-coated super black breast feather of *Ptiloris paradiseus* retains a black appearance indicating structural absorption. SEM stubs are 12.8 mm in diameter. See Fig. 1 for inset photo credits

type's response to the spectrum of an adjacent "white" point—usually a specular highlight[29]. Reducing the value of the denominator in this correction to zero would effectively eliminate the individual's capacity for color correction.

Further research is required to understand the role of multiple scattering among barbs and barbules of multiple feathers in structural absorption by the entire plumage, and on the color correction mechanisms of birds. However, it is clear that structural absorption should be considered along with pigments, structural coloration, and specular reflection, as an important component in determining the visual appearance of organisms. Biological examples of structural absorption have in at least one case inspired the fabrication of new biomimetic materials[15], and the feather structures described herein may have similar direct applications.

## Methods

**Specimens**. Five bird species with profoundly black plumage and two species with normal black plumage were identified by visual observation of museum study skins from the Yale Peabody Museum (YPM), Harvard Museum of Comparative Zoology (MCZ), American Museum of Natural History (AMNH), and the University of Kansas Biodiversity Institute (KU). Details of the specimens and plumage patches studied are summarized in Supplementary Table 1. To the human observer, super black plumage had a strongly matte appearance with so little specular reflectance that it was difficult to focus on the surface of the plumage and distinguish individual feathers. The species with normal black plumage lacked any conspicuous glossy specular highlights. Individual contour feathers were sampled

from museum skins for scanning electron microscopy (SEM) and synchrotron-radiation X-ray microtomograhy (nano-CT). We could not obtain SEM of *Lophorina superba* back feathers or CT scans for *Lophorina superba* back and display cape feathers due to availability of material. Visual inspection of the *Lophorina* back plumage using a light microscope confirmed that the barbules have normal morphology, without the modified barbule arrays present in super black feathers.

**Spectrophotometry**. Light reflectance and absorbance by the plumage can be influenced by the specific orientation of the feathers in the plumage and also by the interaction of light scattered by multiple feathers. The optical properties of the intact plumage cannot be reconstructed reliably by plucking feathers and then laying them (singly or together) on a different surface. Therefore, reflectance spectra of super black and normal black plumage patches were recorded directly from the plumage of prepared museum skins.

Total integrated (diffuse and specular) reflectance spectra were measured with an Ocean Optics USB2000 spectrophotometer and ISP-REF integrating sphere using a Spectralon white standard (Ocean Optics, Dunedin, FL). The light source provided diffuse light from all directions and the gloss trap was closed to collect both specular and diffuse reflectance. To ensure repeatable measures of reflectance from these profoundly black samples, we averaged 10 scans for each output file, and used an integration time of 40 μs. For each patch, we measured three spectra from three different positions within the patch and averaged them to produce a single spectrum for the patch. Two specimens per species were measured for all species except for *Astrapia stephaniae* and *Parotia wahnesi*, for which only one specimen was measured due to availability of material.

Directional reflectance spectra were measured with an Ocean Optics USB2000 spectrophotometer and Ocean Optics DH-2000Bal deuterium–halogen light source (Ocean Optics, Dunedin, FL, USA). The geometry of the directional reflectance measurements placed the detector at 0° normal to the plumage, which

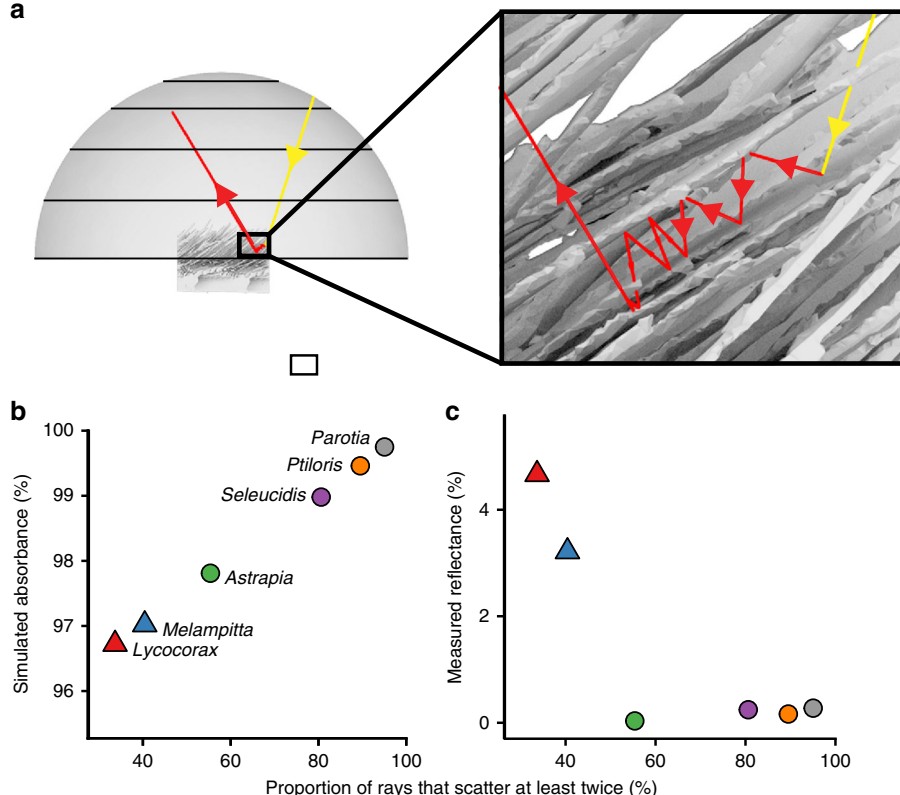

**Fig. 4** Ray-tracing simulations. **a** Simulation from FRED showing the trace of a ray that scatters multiple times between barbules of a super black feather. **b** Substantial variation in frequency of multiple scattering events among species predicts variation in structural absorption. **c** Measured reflectance is significantly negatively correlated with the proportion of reflected rays that scattered at least twice (linear regression: $R^2 = 0.68$, slope = −0.063, SE = 0.021, and $P < 0.05$)

would be the specular direction for typically flat materials. A bifurcated illumination/detection optical fiber was held in an anodized aluminum block ~6 mm above and perpendicular to the plumage. A ~3-mm-diameter circle of light illuminated the plumage. Reflectance between 300 and 700 nm was recorded to obtain the species spectra for the patch. Measures of super black plumage reflectance were quite low and noisy, and signal processing was required. Negative values were converted to 0, and five spectra from each individual were averaged to produce an average spectrum for the patch. Loess smoothing was applied to produce a reflectance spectrum curve (Supplementary Fig. 2).

The light source in our integrating sphere lacked near-ultraviolet light (300–400 nm), but the directional reflectance measures confirmed that none of these patches produced UV reflectance features. Reflectance, %R, was calculated as the area under the measured reflectance spectrum between 400 and 700 nm using Riemann sums and was normalized by the number of wavelength bins measured and 100% reflectance of the white standard.

**SEM.** For SEM, feathers were mounted on stubs using carbon-adhesive tabs, coated with ~15 nm of gold, and viewed and micrographed using an ISI SS40 SEM operating at 10 kV. For *Parotia wahnesi*, *Ptiloris paradiseus* (Fig. 3d), and *Melampitta lugubris* (Fig. 3c) feathers were coated with 5 nm of gold, and then viewed and photographed using a SEM-4 FESEM Ultra55 operating at 5 kV.

**Nano-CT.** For nano-CT, one black contour feather from each species was washed and then soaked in an aqueous solution of Lugol's solution—1% (wt/v) iodine metal ($I_2$) + 2% potassium iodide (KI) in water—for 2–3 weeks to improve X-ray contrast[30]. Feathers were scanned at beamline 2-BM at the Advanced Photon Source facility at U.S. Department of Energy's Argonne National Laboratory, Argonne, Il. Feathers were mounted to a post using modeling clay and surrounded by a Kapton tube to reduce sample motion. Feathers were aligned in the beam to scan a portion of the distal tip that is exposed in the plumage. Scans were made with an exposure time of 30 ms at 24.9 keV to acquire 1500 projections as the sample rotated 180° at 3° s⁻¹. Data sets were reconstructed as TIFF image stacks using the TomoPy Python package (https://tomopy.readthedocs.io) in Linux on a Dell Precision T7610 workstation with two Intel Xeon processors yielding 16 cores, 192-GB RAM, and NVIDIA Quadro K6000 with 12-GB VRAM. The isotropic voxel dimensions of the image stacks were 0.65 μm and the field of view of each data set was ~1.5 mm³.

**3D polygon models.** The external surface of each feather was segmented in VGStudioMAX 2.0 (Volume Graphics) and a 3D polygonal mesh comprising a geometric model of the external surface was extracted using the QuickMesh setting and exported as an OBJ file. To optimize the ray-tracing simulations, each polygonal model was cropped to a 500-μm by 500-μm swatch of the feather vane and then the triangle count was further reduced using the decimate feature (tolerance set to 325 nm) in Geomagic Wrap (3D Systems). Finally, we used the Mesh Doctor feature in Geomagic Wrap to make the surface model manifold, i.e., "water tight." This last step was necessary to repair any defects in the polygonal mesh through which simulated rays could artifactually enter and become trapped inside the feather during ray-tracing simulations.

**Ray-tracing simulations.** The directional reflectance, transmittance, and absorbance of super black and normal black plumage patches were analyzed by numerical ray trace simulations using the software package FRED[31] (Photon Engineering LLC). Simulations employing two types of illumination were conducted for each feather: (1) omni-directional and (2) directional.

The "omni-directional" setup was configured with a hemispherical light source, a hemispherical reflectance detector, and a hemispherical transmittance detector. Into this setup, we imported a 3D polygonal mesh of each feather. Feathers were placed at the center of all three hemispheres and oriented with their vanes in plane with the base of the hemispheres and perpendicular to their poles. The upper or obverse feather surface was oriented toward the light source and reflectance hemisphere; the lower or reverse feather surface was oriented toward the transmittance hemisphere. One million rays of random wavelength between 300 and 700 nm were emitted from random positions on the hemispherical source and propagated in random directions constrained by a square plane with a side length of 330 μm centered on the feather (corresponding to 66% of the width of the feather swatch).

The "directional" setup was configured as a scale model of the directional reflectance spectrophotometry setup. Directional reflectance simulations were conducted for each feather sample under three different light source orientations: (i) tilted +45° toward the proximal end of the feather, (ii) 0° normal to the feather, and (iii) tilted −45° toward the distal end of the feather. While the plumage was illuminated by a 3-mm-diameter spot in the spectrophotometry experiments, the illumination spot in the simulations was scaled from 11% to 330 μm (corresponding to 66% of the 500-μm width of the feather swatch). The width of the light source representing the bare optical fiber bundle and its distance above the

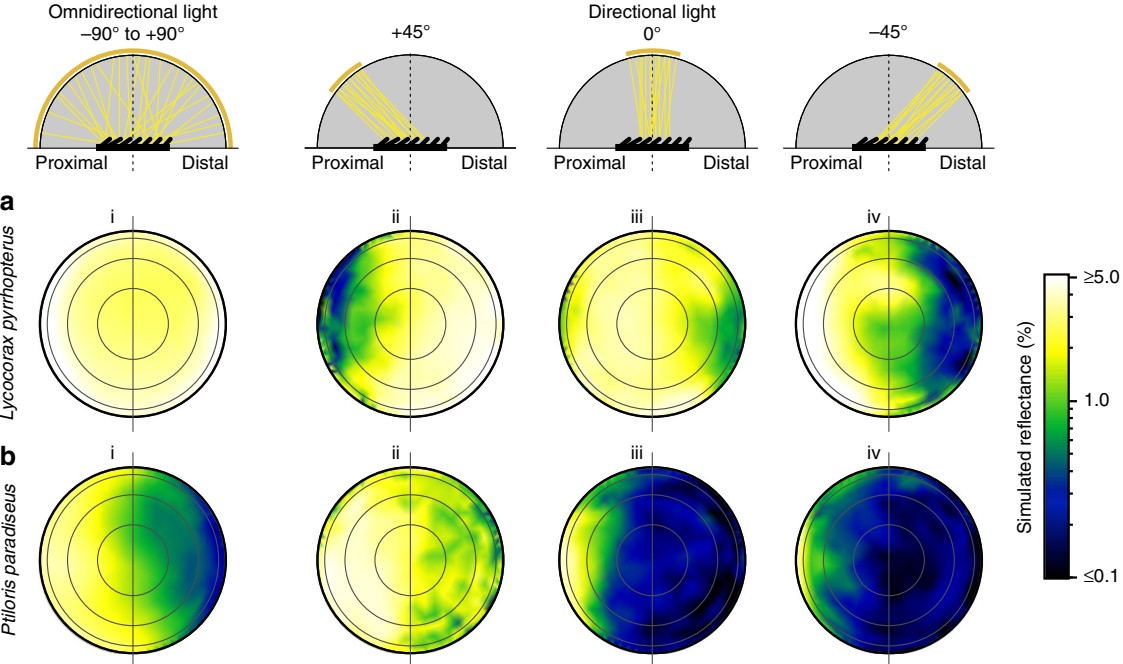

**Fig. 5** The impact of viewing direction and illumination on the appearance of feathers. **a** Normal black *Lycocorax* back feather. **b** Super black *Ptiloris* breast feather with modified barbule arrays. Directional reflectance from ray-tracing simulations is plotted as a log-scale color gradient on orthogonal projections of the reflectance hemisphere under four different lighting conditions (i–iv). Concentric rings represent 22.5°, 45°, 67.5°, and 90°. Horizontal line separates proximal and distal viewing quadrants. Schematics show a cross-sectional view of the optical setup of each illumination. Reflectance hemisphere in gray, feather in black. Concentric gold band indicates the angle of light source, yellow lines show the path of incident light (subsequent path of reflected light not shown). For normal black *Lycocorax*, the darkest viewing quadrant varies with the direction of incident light (compare ii with iv), whereas super black *Ptiloris* shows a strong directional bias in which the distal viewing quadrant is the darkest under all lighting conditions. See Supplementary Fig. 9 for additional species

feather swatch were also scaled at 11% to ensure that the size of the solid angle illuminating the plumage patch in the simulations matched that in the spectrophotometry measurements. One million rays of random wavelength between 300 and 700 nm were positioned on a grid spanning the light source. A circular aperture was used to cull rays from the square source, thereby shaping the source to match that of the spectrophotometer probe. Ultimately, 785,398 rays were emitted by the circular source in random directions within an angular range of 28°, thereby illuminating the 330-μm-diameter spot centered on the feather.

In both "omni-directional" and "directional" simulations, each ray had one of three possible fates. (1) No interaction, where the ray passes through gaps in the feather vane without ever striking the surface of the feather and ultimately terminates when it intersects the transmittance hemisphere. (2) "Transmitted," where the ray strikes the surface of the feather one or more times until it ultimately exits the underside or reverse surface of the feather vane and terminates on the transmittance hemisphere. (3) "Reflected," where the ray strikes the surface of the feather one or more times until it ultimately exits the topside or obverse surface of the feather vane and terminates on the reflectance hemisphere. For the scope of this study, we only consider the subset of incident rays that are "reflected" (fate 3). Rays that terminate on the transmittance hemisphere (fates 1 and 2), represent more complex interactions between multiple overlapping feathers in the plumage and/or the skin that we do not consider here.

We simplified the ray-tracing simulations of the feather surface and controlled for potential differences in surface roughness between the real feathers by excluding surface scattering caused by surface roughness (reflections in nonspecular directions) from the simulation. We traced rays using the surface normals of the bare polygon mesh of the feather, treating each polygon in the mesh as a smooth surface. Since no BRDF model was applied to the surface, all radiant power was directed in the specular direction. Thus, each time a ray struck the surface of the feather (a simplified "scattering" event), it bifurcated into one and only one component ray that reflected from the surface of the feather, and one and only one component ray that transmitted into the feather. The direction of the reflected ray was computed based on the law of reflection ($\theta_i = \theta_r$), and Fresnel equations yielded the fraction of the incident radiant power reflected as a function of the incident angle and the ratio of the index of refraction of air (1.0) and feather keratin (1.56). To investigate the effects of surface microstructure independent of any potential differences in melanin or internal nanostructure between the real feathers, we assumed that rays transmitted into the feather were entirely absorbed before exiting the feather. Thus, any difference in calculated absorption between simulated feathers is caused by variations in the orientation of the feather surface and differences in the number of multiple scattering events.

The ray-tracing simulation proceeded as follows: first, rays with equal amounts of radiant power were emitted from the light source and propagated in the direction of the feather. Then, rays repeatedly intersected the surfaces of the feather vane and reflected from those surfaces in the specular direction until they exited the volume of space occupied by the feather vane and terminated on a hemisphere. For each ray, the simulation recorded the number of light ray-surface intersections, the hemisphere of and spherical coordinates of the termination point, and the ending radiant power. For each ray, absorbance was calculated from the difference between the starting and ending radiant power. For comparison with the directional reflectance spectrophotometry measurements, total absorbance under 0° normal directional illumination was calculated as the sum of reflected light rays that terminated within an angular range of 27°. Percent multiple scattering was calculated as the percentage of this set of rays that scattered two or more times off of the surface of the feather.

To determine how reflectance varies based on the angle incident light and viewing directions, we calculated the locally averaged reflectance at different viewing directions with a nonparametric kernel regression fit using the kreg function with default settings from the R package gplm. The kernel density estimate and regression fits were evaluated at 400 points, representing different viewing directions that were uniformly distributed over the reflectance hemisphere, and the results were plotted as a log-scale color gradient on orthogonal projections of the hemisphere using the persp3d function from the R package rgl.

We used linear regression to estimate correlation between the proportion of rays that scattered at least twice and the actual measured reflectance for the 0° normal directional light ray-tracing setup; we report $R^2$, slope, standard error of the slope, and P value.

**Data availability**. The data that support the findings of this study are available from the corresponding author on request.

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

## Acknowledgements

Xianghui Xiao and the 2-BM beamline group provided assistance with CT scanning and analysis at the APS facility, Argonne National Lab. Jeremiah Trimble and Kate Eldridge assisted with collections usage. The research has been improved by discussions with David Brainard, David Shoehalter, Kristof Zyskowski, David Haig, James Harvey, Ryan Irvin, and members of the Prum Lab. Ed Scholes, Tim Laman, Trans Niugini Tours, Hanom Bashari, and Daniel López kindly gave permission to reproduce photos of birds of paradise (Fig. 1). Andrew Farris assisted with kermal regressions for the directional reflectance plots. This research was funded by the W. R. Coe Fund of Yale University, by a Sigma XI student research fellowship to D.E.M., and by a Mind, Brain, and Behavior Graduate Student Award to D.E.M. D.E.M. was supported by the Department of Defense (DoD) through the National Defense Science and Engineering Graduate Fellowship (NDSEG) Program. D.E.M. was also supported by a Theodore H. Ashford Graduate Fellowship in the Sciences. Tomography data collections at the Advanced Photon Source beamline 2-BM, Argonne National Laboratory were supported by the U.S. Department of Energy Office of Science (Proposal ID 41887). T.J.F. was supported by a NSF Post-doctoral Fellowship in Biology (#1523857). Richard Pfisterer of Photon Engineering graciously licensed FRED to T.A.H. for this research. This work was performed in part at the Harvard University Center for Nanoscale Systems (CNS), a member of the National Nanotechnology Coordinated Infrastructure Network (NNCI), which is supported by the National Science Foundation under NSF ECCS award no. 1541959.

## Author contributions

All authors conceived the research design, analyzed the data, and wrote the paper jointly. D.E.M. and R.O.P. performed the spectrophotometry, and D.E.M. conducted the SEM experiments. T.A.H. performed the ray-tracing experiments. T.A.H., T.F., and R.O.P. performed CT scans and analyzed ray-tracing experiments. D.E.M., T.A.H., and T.F. prepared the figures.

## Additional information

**Competing interests:** The authors declare no competing financial interests.

