## [Peer Review File · Nature Communications]

Reviewers' comments:

Reviewer #1 (Remarks to the Author):

This work concerns a quantitative characterization of structurally assisted absorbance in the feathers of several species of birds - and in particular in areas of birds of paradise that are adjacent to patches of structural colouration, areas that act as a highly conspicuous visual signal.

Overall this is one of the best papers I have read in a long time. The methodology is exact and the right mix to characterise the mechanisms under investigation.

I only have two minor comments and would ask the authors to include just a few lines in the paper to provide this extra context. Otherwise, the paper should be published.

1) This sort of low reflectance parallels that seen in other animals - for example in butterflies. The context of the work would be improved if the authors could discuss any similarities or differences between the structural mechanisms.

2) The quantification is very well discussed in the paper, however, it is very directed towards human measures and quantities. What difference does a reduction of reflectance of 5% make to the intended receiver? How does a visual distance change in a tetrachromatic colour space? I ask the authors to expand some of the discussion here and put the hypothesis in the context of the bird's visual systems and discuss and assess the effect of noise in cone photoreceptors. Does a reduction in the reflectance improve the SNR in the contrast? If the normal black is close to the noise floor, what difference does the extra reduction make?

Reviewer #2 (Remarks to the Author):

The authors use tomography and ray simulations to identify the mechanisms of "super black" color production in some birds of paradise. The authors have taken an interesting, interdisciplinary approach and I appreciate the amount of work they have put into this question. The data convincingly show that some of the blackness of these feathers is produced by a non-pigmentary, i.e. structural mechanism. The ray-tracing experiment and the flatness of the reflectance curves indicate a contribution of feather structure to black color production.

However, the results and their novelty are oversold. A mean reflectance of ~4% is not "superblack"- it is only marginally blacker than the presented control black feathers, is well within the range of other black feathers and is quite far off from true superblack surfaces such as Vantablack (~0.04% reflectance). The flatness of the curve is interesting, but would hardly qualify the material as superblack. The authors need to shift the focus of the paper away from these distracting hyperbolic claims and towards the interesting, but perhaps lower impact, discovery of structurally assisted blackness in feathers. This finding is not entirely novel (see e.g. Vukusic et al. for butterflies), but is new for birds.

That being said, the authors have not adequately explored a critical component that may explain why their objective measurements don't appear to adequately represent the observed color of these feathers, and may also provide insight into the mechanism. Specifically, these feathers appear to have extremely low specular reflectance. Indeed, the authors state that they had difficulty measuring it because it was so low. But they do not report these results nor those from "control" black feathers. I am nearly certain that control feathers will have high specular reflectance relative to the BOP feathers, and that this difference will be much more obvious than the small one between their diffuse reflectances. Lower specular reflectance, and not virtually identical diffuse reflectance, is likely to be the main cause of the blacker appearance of BOP feathers.

This is critical because it also likely helps to explain the mechanism for this difference. While the barbule structure may indeed cause somewhat lower diffuse reflectance, it also creates an extremely rough surface that likely lowers specular reflectance. Simply comparing surface roughness of the black feathers will be an important and necessary addition to the paper.

L43: None of the papers cited here or on L51-52 describe their materials as super black, or as anything other than "black." The authors of the current paper should follow their lead.

L64: here the authors note that the BOP feathers lack specular highlights. It is strange that they do not follow up on this critical observation to a greater extent.

L77: Again, none of these papers refer to these materials as super black. The authors should read and cite some of the literature on true "super black" materials (e.g. Brown et al. 2002. The physical and chemical properties of electroless nickel-phosphorus alloys and low reflectance nickel-phosphorus black surfaces. *J. Mater. Chem.*, 2002,12, 2749-2754. These materials reflect <0.04% of light, an order of magnitude less than the BOP feathers. In any case, this statement that both normal black and BOP black feathers were within the same range does not support their consistent claims of superblackness.

L108: I'm surprised that the authors did not perform TEM or even SEM of a cross section of these feathers. Perhaps there is an important underlying nanostructure in the melanin?

L109: this is a neat experiment

L140: This is incorrect. Only one normal black feather (*Lycocorax*) had <50% of rays scattered at least twice.

Minor note: please use consistent terminology: either "more than once" or "at least twice", not both.

L177: The authors again discuss how critical specular reflectance is, but do not address it in their experiments.

L186: Many studies have shown effects of macrostructure on color, see e.g. references 3 (manmade) and 7 (natural) here or Shawkey and D'Alba 2017 Interactions between color-producing mechanisms and their effects on the integumentary color palette. *Philosophical Transactions of the Royal Society of London B* 372: 20160536 for more natural examples. This should be reworded to make this clear.

L201: Is this correct that "normal black plumage lacked specular highlights", or did you mean "had specular highlights."

L209: Plucking feathers and measuring them separately in stacks of identical numbers allows for control of the effects of variation in number of feathers (at the expense of biological realism). What if the Bop patches were just blacker because they contained more

feathers?

L224: These low specular reflectance data are critical. It is not clear what the authors mean by "repeatable"- sometimes they were 10%, other times 0%? Or something else? They should at least show some of these measurements. Moreover, they should compare them to those from normal black feathers to illustrate and (perhaps) quantify the differences in specular reflectance.

L292: Can the authors please justify modeling these as mirrors when they are clearly far from being mirrors?

Reviewer #3 (Remarks to the Author):

In this manuscript, the authors identify modified feather structures in birds of paradise that, through structural absorption, result in "super black" colors. They combine spectrophotometry, mathematical models, electron microscopy and nanotomography to show that these modified structures are responsible for the observed effect. The sampling and modeling seems well-thought and the results robust, so I find myself pleasantly surprised with little to comment on!

My main (yet still minor) comment is that some of the assertions (e.g. line 78-96) are a bit too strong given the lack of statistical comparison, quantification of variation and of the magnitude of differences given this natural variation (which is understandable given the complexity of measurements taken and the rarity of species where this phenomenon is observed - so this isn't meant as a criticism of sampling, but an acknowledgment of its limitations). I would suggest a more speculative and hypothetical language use in this section.

I also think it is very important to include a measure of variation in figure 1C for a more transparent presentation of the differences being suggested - maybe given the small sample size it would be adequate to present the average curves for all species/patches, perhaps in a lighter weight line, with the average per group (normal/superblack) in a thicker weight line.

In Lines 25, 136, 139, what is the "SE" value reported? is it the standard error of the regression slope? If so, present the regression slope value, as well as the t-value from which the p-value was obtained. It's not really clear what analysis was conducted here, if it is a correlation or a regression (I'm assuming the latter based on the statistics presented), I couldn't find any mention of statistical analyses in the methods. Why is the adjusted R^2 being used, were any covariates added to the model?

Extended figure 2 would benefit from more appropriate y-axis limits (none of the spectra go above 10 or so)

Extended table 3 seems to have redundant footnotes and headers

Response to Reviewers

Revised Manuscript

“Structural Absorption by Barbule Microstructures of Super Black Bird of Paradise Feathers”

We would like to thank the reviewers for their constructive comments and corrections. In light of their suggestions, we have included new experimental data and analyses and have made substantial revisions to the manuscript. In particular, Reviewer #2 raised concerns that we had not fully characterized the mechanism for structurally assisted blackness in feathers. To address these concerns, we conducted additional spectrophotometry measurements and ray-tracing simulations. We have incorporated an additional set of directional reflectance measurements of all feathers using a second spectrophotometry experimental setup to complement the original total integrated (specular and diffuse) reflectance measurements. Furthermore we have conducted 18 new ray tracing simulations (3 new experimental setups each for 6 feathers) to complement the original 6 simulations. Our primary result holds, and is now even better supported: super black feathers from birds of paradise structurally absorb light through multiple scattering. Indeed, the new directional reflectance measures demonstrate that super black bird of paradise feathers reflect as little as 0.05%-0.31% of incident light. This is one or two orders of magnitude less than black plumage of closely related birds and comparable to man-made ultra-absorbent, super-black materials. The additional experimental setups have also allowed us to more fully characterize the unusual directional reflectance bias in super black feathers. We have made substantial revisions to the text and figures of the manuscript to incorporate our new results and conclusions.

Please see below for our detailed response to reviewers.

Reviewer 1

- 1) *“This sort of low reflectance parallels that seen in other animals - for example in butterflies. The context of the work would be improved if the authors could discuss any similarities or differences between the structural mechanisms.”*

Response: We now include a comparison of reflectance curves and qualitative appearance of structurally absorbing materials between these animals in the spectrophotometry results. We have added an additional discussion to the conclusion noting that the feathers, snake scales, and butterflies represent 3 distinct morphologies revealing a diversity of structural mechanisms in nature.

- 2) Broad request to discuss reflectance less in terms of humans and more in terms of the birds
 - a) *‘What difference does a reductions of reflectance of 5% make to the intended receiver?’*
Response: Our new, additional data analyses show that for biologically relevant viewing directions- that is, the head-on viewing of males by females—the difference between normal black and super black is one or two orders of magnitude. In goldfish and humans, perceptual experiments confirm that changes in blackness of background do have behaviorally significant effects.
 - b) *‘How does a visual distance change in a tetrachromatic colour space?’*

Response: Visual distance concerns spatial acuity, and does not affect chromatic or

achromatic perception captured by color space.

- c) *'I ask the authors to expand some of the discussion here and put the hypothesis in the context of the bird's visual systems and discuss and assess the effect of noise in cone photoreceptors. Does a reduction in the reflectance improve the SNR in the contrast?'*

Response: The interaction between the reflectance of the black patch and the signal provided by the adjacent colorful patch is complex. The contrast is not merely between patches, but between the signal and the bird's estimate of the illumination of the signal. We propose that the reduction in reflectance affects the individual's estimate of the quantity (and possibly color) of the ambient light illumination of the signal. Theoretically, white balancing (von Kries transformation) involves dividing each cone stimulus by the cone response to the spectrum of a "white" point. Reducing the value of this denominator in this correction to zero essentially eliminates the capacity of for color correction. We added text to this effect in the manuscript.

- d) *'If the normal black is close to the noise floor, what difference does the extra reduction make?'*

Response: In goldfish and humans, perceptual experiments confirm that changes in blackness of background do have visually significant effects. It is beyond the scope of the current study to quantitatively assess differences in behavior that correspond to quantized differences in background luminance. Our new, data analysis shows that for biologically relevant viewing directions- that is, the head-on viewing of males by females—the difference between normal black and super black is one or two orders of magnitude. The discovery of unique feather morphologies that achieve these striking optical effects is good evidence that these differences in reflectance do matter to the organisms. But that is the subject of future research.

Reviewer 2

- 1) Comments concerning the term 'superblack'

- a) *"However, the results and their novelty are oversold. A mean reflectance of ~4% is not "superblack"- it is only marginally blacker than the presented control black feathers, is well within the range of other black feathers and is quite far off from true superblack surfaces such as Vantablack (~0.04% reflectance). The flatness of the curve is interesting, but would hardly qualify the material as superblack. The authors need to shift the focus of the paper away from these distracting hyperbolic claims and towards the interesting, but perhaps lower impact, discovery of structurally assisted blackness in feathers. This finding is not entirely novel (see e.g. Vukusic et al. for butterflies), but is new for birds.'*

Response: At this reviewer's suggestion, we took additional spectrophotometry measurements and found 1-2 orders of magnitude difference between control and super black feathers in normal, directional reflectance (see Figure 2, Extended Table 1); *Astrapia* actually approaches Vantablack; this bird has 0.05% specular reflectance, and the other super black birds range between 0.24% and 0.4% versus >3% for control black feathers. This large difference in reflection, as demonstrated by our simulations, is attributable to structural absorption. In addition, the flatness of the curve is in fact a

hallmark of super black materials: in Panagiotopoulos et al. 2012 ("Nanocomposite catalysts producing durable, super-black carbon nanotube systems: applications in solar thermal harvesting." ACS nano 6.12 (2012): 10475-10485.), the authors note that their engineered material has exceptionally low reflection "in the entirety of the visible range." This is a desired characteristic of many engineered materials, particularly those engineered to efficiently harvest solar thermal energy.

- b) L43: None of the papers cited here or on L51-52 describe their materials as super black, or as anything other than "black." The authors of the current paper should follow their lead.

Response: We borrowed this term from biomimetic materials science; see, for example: Zhao et al 2011 "**Super black** and ultrathin amorphous carbon film inspired by anti-reflection architecture in butterfly wing." Panagiotopoulos, Nikolaos T., et al. "Nanocomposite catalysts producing durable, **super-black** carbon nanotube systems: applications in solar thermal harvesting." ACS nano 6.12 (2012): 10475-10485. One feature of super black materials is their broadband absorption; as the authors write in Panagiotopoulos et al (2012), their materials has exceptionally low optical reflection "in the entirety of the visible range." As a new convention for super black bio-materials, we suggest in this revision that they be defined as having <0.5% reflectance across the relevant visible spectrum.

- c) L77: Again, none of these papers refer to these materials as super black. The authors should read and cite some of the literature on true "super black" materials (e.g. Brown et al. 2002. The physical and chemical properties of electroless nickel–phosphorus alloys and low reflectance nickel–phosphorus black surfaces. J. Mater. Chem., 2002,12, 2749-2754. These materials reflect <0.04% of light, an order of magnitude less than the BOP feathers. In any case, this statement that both normal black and BOP black feathers were within the same range does not support their consistent claims of superblackness.

Response: See previous two responses; with new measurements, our feathers are in the appropriate range; further, the term super black has been used in materials science as cited above. In the paper suggested here, Brown et al. 2002, the materials are ~0.4% reflectance, which is equaled by our super black feathers.

2) Comments concerning specular reflectance

- a) *'That being said, the authors have not adequately explored a critical component that may explain why their objective measurements don't appear to adequately represent the observed color of these feathers, and may also provide insight into the mechanism. Specifically, these feathers appear to have extremely low specular reflectance. Indeed, the authors state that they had difficulty measuring it because it was so low. But they do not report these results nor those from "control" black feathers. I am nearly certain that control feathers will have high specular reflectance relative to the BOP feathers, and that this difference will be much more obvious than the small one between their diffuse reflectances. Lower specular reflectance, and not virtually identical diffuse reflectance, is likely to be the main cause of the blacker appearance of BOP feathers. This is critical because it also likely helps to explain the mechanism for this difference. While the barbule structure may indeed cause somewhat lower diffuse reflectance, it also creates*

an extremely rough surface that likely lowers specular reflectance. Simply comparing surface roughness of the black feathers will be an important and necessary addition to the paper.'

Response: The reviewer raises the concern here and elsewhere that our original spectrophotometry measurements only report the difference in diffuse reflectance between feathers and thus we are lacking important measures of specular reflectance. We would like to clarify that we did measure specular reflectance in our original submission but failed to clearly describe our experimental setup. The original integrating sphere measurements were done in such a way as to capture total specular and diffuse reflectance under diffuse lighting. Specifically the gloss trap on the integrating sphere was closed during all measurements to include specular reflectance. So these measures include all of specular reflectances together. Nevertheless, we now include spectrophotometry measures of reflectance using a normal (0°) directional light source and detector. These are the conditions that will maximize specular reflectance from a flat surface. Both experimental setups demonstrate that super black feathers do have lower specular reflectance than normal black feathers. Our ray tracing simulations (which were originally designed to explicitly control for any potential differences in surface roughness) demonstrate that the modified barbule structure causes multiple scattering of light which results in lower specular reflection through structural absorption.

- b) L64: here the authors note that the BOP feathers lack specular highlights. It is strange that they do not follow up on this critical observation to a greater extent.

Response: Because our integration sphere measurements in our original paper include specular reflectance in all directions, our data were directly relevant to this question. However, in this revision, the new normal, directional measurements of specular reflectance, and our simulations to address the reviewers concerns specifically, and add further support for our hypothesis that the barbule morphology of super black plumage effectively eliminates both specular and diffuse reflectance in directions we hypothesize are critical to effective courtship display.

- c) L177: The authors again discuss how critical specular reflectance is, but do not address it in their experiments.

Response: See 2b.

- d) L201: Is this correct that “normal black plumage lacked specular highlights”, or did you mean “had specular highlights.”

Response: Indeed, the normal black plumage of our study skins did not produce obvious specular highlights visible to the human observer, i.e. the plumage does not have a glossy, specular appearance.

- e) L224: These low specular reflectance data are critical. It is not clear what the authors mean by “repeatable”- sometimes they were 10%, other times 0%? Or something else? They should at least show some of these measurements. Moreover, they should compare them to those from normal black feathers to illustrate and (perhaps) quantify the differences in specular reflectance.

Response: When we originally took the measurements, we did not report them because in each measurement there were multiple negative values reported and overall extremely

low reflectances (<0.5%). We considered them within the range of instrument noise. At the suggestion of this reviewer, we retook these measurements and found that they were actually repeatable and significant. As reported in our revision, we converted artifactual negative values to 0%, performed a Loess smoothing, and report these data herein (Figure 2, Extended Table 1). The super black feathers are 1-2 orders of magnitude darker than control black feathers, and approach the extremely low reflectance of man-made surfaces such as Vantablack.

- 3) Comments concerning other aspects that might be contributing to differences in blackness.
General Response: Reviewer 2 raises several points concerning other aspects of the feathers and that could be contributing to the observed differences in reflectance within our sample of feathers. This includes the internal nanostructure, surface roughness, and density of feathers in the plumage. As has been demonstrated by previous research, all of these additional aspects of the feather could also be contributing to the overall variation in blackness between our feathers. In general, we did not empirically investigate these other features because we did not intend for our paper to provide a comprehensive description of all of the sources contributing to the observed variation in reflectance within our sample of feathers. Instead, the primary aim of our paper is to demonstrate, for the first time, barbule morphologies cause multiple scattering of light, leading to structural absorption, and that this mechanism contributes to (but is not necessarily exclusively responsible for) the observed variation in reflectance between feathers. To that end, our ray tracing experiments were designed to specifically control for any potential differences between the feathers in internal melanin, nanostructure, surface roughness (external nanostructure), and feather density (see responses below for details). So, we have demonstrated that these microscale morphologies can produce the observed differences in reflectance by structural absorption.

- a) L108: I'm surprised that the authors did not perform TEM or even SEM of a cross section of these feathers. Perhaps there is an important underlying nanostructure in the melanin?

Response: There very well could also be a difference in the underlying nanostructure in the melanin but this was beyond the original scope of our paper. We specifically controlled for any potentially differences in melanin nanostructure between the feathers in our ray tracing simulations by assuming that 100% of the light transmitted into the feather was 100% absorbed before exiting the feather. In other words the ray tracing experiments effectively assumed that all feathers had the same internal nanostructure.

- b) *While the barbule structure may indeed cause somewhat lower diffuse reflectance, it also creates an extremely rough surface that likely lowers specular reflectance. Simply comparing surface roughness of the black feathers will be an important and necessary addition to the paper.'*

Response: The literature on surface roughness is entirely framed as variations or perturbations from flatness. While this could easily be applied to the normal black feather barbules, it is not really the appropriate spatial scale to ask the question about the light scattering behavior of the extraordinarily complex surfaces of the birds of paradise feathers. However our tomographic models provide much better physical models of the feather surfaces than a greatly simplified and mostly inappropriate mathematical index of surface roughness. In our scattering simulations, we assumed a smooth surfaces for both

sets of feathers at the scale of the size of light wavelengths. From visual inspection of SEMs, there was no observable difference in barbule surface roughness between control and super black feathers. We report all SEMS in our extended materials. For this reason, we chose not to (i) report surface roughness nor (ii) apply a BRDF function to our simulation. It would not have been possible to apply a realistic function, and we chose to isolate the striking variable of micro-scale barbule morphology. Lastly, it is possible that one could use the term roughness (“it also creates an extremely rough surface”) to refer to exactly these micro-scale variations created by the barbule morphology, but the term roughness is more generally used for ~150nm variations in the surface of an object. Our model specifically accounted for this large scale “barbule roughness” while controlling for traditional, ~100nm scale surface roughness.

- c) L292: Can the authors please justify modeling these as mirrors when they are clearly far from being mirrors?

Response: We modeled the feather surface as an ideal mirror (that is, at the size scale of the wavelengths of light) for two reasons. First, this allowed us to specifically control for any potential differences in surface roughness between the real feathers by effectively assuming they all had the same roughness (i.e. none). Second, we chose to model the feathers as a mirror (as opposed to specifying some value of roughness) because it greatly simplifies the experiment by restricting all light to only specular reflection. Therefore, any differences in brightness predicted by the ray tracing simulations are directly a result of the microscale multiple scattering events that we can simulate directly with the tomographic models of the surfaces of the feathers. We have updated the text of our manuscript to clarify our justifications.

- d) L209: Plucking feathers and measuring them separately in stacks of identical numbers allows for control of the effects of variation in number of feathers (at the expense of biological realism). What if the Bop patches were just blacker because they contained more feathers?

Response: Again, there very well could also be a difference in the density of feathers within the plumage patch that contributes to the observed difference in reflectance in the real birds but this is beyond the scope of our study. We specifically controlled for any potential differences in the density of feathers in plumage patches by modeling only a single, isolated feather in our ray tracing experiments. Our analyses of the light scattering behaviors of the tomographic models demonstrate that microscale features can contribute the differences in reflectance. Also, our observation that individual gold coated super black feathers retain their black appearance while a normal black feathers turn gold is physical proof that individual feather structure is sufficient to explain much of this effect in some plumages (Figs 3C,D). Lastly, we note that portions of the super black 'cape' of *Lophorina superba* are only a single feather thick when the cape is opened.

4) Other miscellaneous comments

- a) L109: this is a neat experiment

Response: Agreed!

- b) L140: This is incorrect. Only one normal black feather (*Lycocorax*) had <50% of rays scattered at least twice. Minor note: please use consistent terminology: either “more than once” or “at least twice”, not both.
Response: The *Lycocorax* statement has been removed and we have edited the text to use consistent terminology
- c) L186: Many studies have shown effects of macrostructure on color, see e.g. references 3 (manmade) and 7 (natural) here or Shawkey and D’Alba 2017 Interactions between color-producing mechanisms and their effects on the integumentary color palette. *Philosophical Transactions of the Royal Society of London B* 372: 20160536 for more natural examples. This should be reworded to make this clear.
Response: We have added citations to these and other papers exploring the effect of macrostructure on color. Our primary contribution, the assessment of structural absorption, has not yet been examined in feathers.

Reviewer 3

- 1) *My main (yet still minor) comment is that some of the assertions (e.g. line 78-96) are a bit too strong given the lack of statistical comparison, quantification of variation and of the magnitude of differences given this natural variation (which is understandable given the complexity of measurements taken and the rarity of species where this phenomenon is observed - so this isn't meant as a criticism of sampling, but an acknowledgment of its limitations). I would suggest a more speculative and hypothetical language use in this section.*
Response: We have edited the text to use more speculative language where appropriate.
- 2) *I also think it is very important to include a measure of variation in figure 1C for a more transparent presentation of the differences being suggested - maybe given the small sample size it would be adequate to present the average curves for all species/patches, perhaps in a lighter weight line, with the average per group (normal/superblack) in a thicker weight line.*
Response: Good point; we replotted the data showing every single species so that the differences were more transparent and easily understandable, and included the actual data points in the plotted spectra.
- 3) *In Lines 25, 136, 139, what is the "SE" value reported? is it the standard error of the regression slope? If so, present the regression slope value, as well as the t-value from which the p-value was obtained. It's not really clear what analysis was conducted here, if it is a correlation or a regression (I'm assuming the latter based on the statistics presented), I couldn't find any mention of statistical analyses in the methods. Why is the adjusted R² being used, were any covariates added to the model?*
Response: No covariates were added to the model. We added explanatory text to the methods: “We used linear regression to estimate correlation between the proportion of rays that scattered at least twice and the actual measured reflectance for the 0° normal directional light ray tracing setup; we report R², slope, standard error of the slope, and p-value..”

- 4) *Extended figure 2 would benefit from more appropriate y-axis limits (none of the spectra go above 10 or so)*

Response: Spectra have been replotted with more appropriate axes.

- 5) *Extended table 3 seems to have redundant footnotes and headers*

Response: Fixed

REVIEWERS' COMMENTS:

Reviewer #1 (Remarks to the Author):

The letter addresses all my comments satisfactorily and the authors have included extra evidence supporting their claims. The manuscript is ready to be published.

Reviewer #2 (Remarks to the Author):

I am glad to see that the reviewers found my suggestions useful. The new data showing extremely low specular reflectance are convincing. It would be nice to see a more concrete statement that this low specular reflectance is the primary cause of the exceptionally dark plumage of these birds. I also note that the authors' definition of "super black" in biological cases as $<0.5\%$ reflectance seems arbitrary and a little.. convenient given that their data are below this self-assigned threshold. I would eliminate the text referring to this definition, as it doesn't help the paper, which is now quite good (although obviously not the last word on the subject), in any substantial way.

Reviewer #3 (Remarks to the Author):

The authors have done a great job addressing previous review comments. A minor visualization suggestion is to use some feature in figure 2 to group spectra in each of the two groups being compared (normal/super black) - such as line type or weight, or annotations. This way the reader can immediately identify the three brighter curves as being the normal ones in both panels (since separation isn't that conspicuous in panel A).

This is a very exciting manuscript, and one that I am looking forward to see in final form.

Regards,
Rafael Maia

RESPONSE TO REVIEWERS:

Reviewer #1 (Remarks to the Author):

The letter addresses all my comments satisfactorily and the authors have included extra evidence supporting their claims. The manuscript is ready to be published.

Thank you.

Reviewer #2 (Remarks to the Author):

I am glad to see that the reviewers found my suggestions useful. The new data showing extremely low specular reflectance are convincing. It would be nice to see a more concrete statement that this low specular reflectance is the primary cause of the exceptionally dark plumage of these birds. I also note that the authors' definition of "super black" in biological cases as $<0.5\%$ reflectance seems arbitrary and a little.. convenient given that their data are below this self-assigned threshold. I would eliminate the text referring to this definition, as it doesn't help the paper, which is now quite good (although obviously not the last word on the subject), in any substantial way.

Thank you; we eliminated the $<0.5\%$ text, and also added a statement about the significant of low specular reflectance.

Reviewer #3 (Remarks to the Author):

The authors have done a great job addressing previous review comments. A minor visualization suggestion is to use some feature in figure 2 to group spectra in each of the two groups being compared (normal/super black) - such as line type or weight, or annotations. This way the reader can immediately identify the three brighter curves as being the normal ones in both panels (since separation isn't that conspicuous in panel A).

This is a very exciting manuscript, and one that I am looking forward to see in final form.

Regards,
Rafael Maia

Thank you; we added dotted lines to help viewers discriminate between the normal and super black feathers.